# The Paradoxical Malthusian. A Promethean Perspective on Vaclav Smil's *Growth: From Microorganisms to Megacities* (MIT Press, 2019) and *Energy and Civilization: A History* (MIT Press, 2017)

**Pierre Desrochers**

Department of Geography, Geomatics and the Environment, University of Toronto Mississauga,
3359 Mississauga Rd., Mississauga, ON L5L 1C6, Canada; pierre.desrochers@utoronto.ca

**Abstract:** Prolific energy writer Vaclav Smil's "Growth: From Microorganisms to Megacities" (MIT Press, Cambridge, MA, USA, 2019) is marketed as the most comprehensive study of the modalities of growth in Earth's life systems in their many natural, social, and technological forms. While the book reflects Smil's strength as a polymath, it also brings into focus his Malthusian outlook. Smil's Malthusianism is puzzling in light of much empirical evidence to the contrary and of his own detailed histories of human technological achievements, including his recent massive synthesis "Energy and Civilization: A History" (MIT Press, Cambridge, MA, USA, 2017). In keeping with Smil's historical emphasis, in this review essay, the Malthusian assumptions, assertions, and conclusions of these books are challenged through the Promethean insights of numerous writers whose output long predates the modern environmental movement and can thus avoid charges of "greenwashing". I make a case that, in the context of market economies (i.e., competition, price system, and private property rights), humans' unique propensity to trade physical goods and to (re)combine things in new ways have long delivered both improved standards of living and environmental remediation. I further suggest that it is not the volume of materials handled, but rather how they are handled that determines the impact of economic growth on the biosphere. While Professor Smil is fond of saying that "numbers don't lie", his work illustrates that they are sometimes made to tell an unduly pessimistic story through the intellectual filters created by an author's assumptions and value judgements.

**Keywords:** Vaclav Smil; promethean; cornucopian; Malthusian; circular economy; decoupling; price system; private property rights

---

*Homines libenter quod volunt credunt*

Men believe what they want

—Publius Terentius Afer (c. 195/185—c. 159? BC), as quoted by Vaclav Smil [1] (p. v)

## 1. Introduction

New books, academic articles, and popular essays on the population, resources, and environment nexus are published every year. Most are intensely partisan, for there can be no middle ground between the beliefs that valuable resources are finite and nature inherently fragile on the one hand, and that human creativity can find ways around scarcity and environmental damage on the other.

So-called "Prometheans" or "Cornucopians" typically argue that transforming the landscape and developing new technologies provide a growing population with increasingly abundant resources and

improved standards of living. This perspective dominated Western thought from the Enlightenment to the rise of the modern environmentalist movement two generations ago. Far from viewing nature as inherently good, Account Prometheans of all political persuasions emphasized its shortcomings, dangers, and resilience. While human actions could create negative consequences, these would be best addressed through further innovations and economic growth [2,3].

By contrast, eco-pessimists have believed since at least antiquity that most non-renewable resources are finite and that human activities are ultimately constrained by the limited carrying capacity of ecosystems [4,5], although these ideas are now typically attributed to Thomas Robert Malthus [6] and William Stanley Jevons [7]. Malthus is remembered (somewhat unfairly) for arguing that food production would not keep up with rapid population growth. Jevons extended the limits reasoning to coal and other valuable commodities. Another concept with deep historical roots that dominated much of the ecological research and environmental policymaking over the twentieth century is that nature is in a harmonious yet fragile balance that shouldn't be disturbed by human activities. Reactionary thinking against industrialization and urbanization, coupled with a long-standing elitist dislike of mass consumption, further shaped modern environmental thought and activism [8–10].

An especially ambitious contribution to this latter perspective was recently published by the Czech-born Canadian environmental scientist and polymath Vaclav Smil. "Growth: From Microorganisms to Megacities" (MIT Press, Cambridge, USA, 2019; henceforth *Growth*) [11] is, as the subtitle implies, an attempt to distil patterns from the study of growth of everything, from archaea to civilizations. While in the hands of almost any other writer this project would reek of hubris, Professor Smil is a throwback to the kind of erudite central European scholar exemplified by eminent social scientists such as Max Weber, Joseph Schumpeter, and Ludwig von Mises whose enormous output was once described in jest as based on the belief that "encyclopedias might very well just vanish from the shelves" [12] (non-paginated, henceforth n.p.). As some readers may know, Professor Smil's breadth and depth of technical knowledge on anything remotely related to energy is truly astonishing. Idiosyncratic, he is a self-described "old-fashioned scientist" who prefers "hard engineering realities" to "interminably vacuous and poorly informed policy 'debates'" [13] (n.p.). His basic outlook is that our economy, societies, civilization, and ultimate survival all depend upon the availability of energy sources, our ability to harness them, and the efficiency with which we are able to convert them into useful things. Always aware of the complexity of human systems, inherent uncertainties, and the limits of human knowledge and understanding, he has throughout his long career steered clear of detailed predictions and policy recommendations. On the other hand, his relative lack of interest in public policy intricacies, casual dismissal of basic economic insights, and often cursory discussions of the impact of incentives and institutions on individual, corporate, and societal behavior have sometimes been deemed problematic [14,15].

Be that as it may, Smil is a man of integrity who, when deriving inescapable conclusions from undisputable facts, has never shied away from criticizing what he deems wishful thinking and energy policy snake oil, along with their delusional or dishonest peddlers (e.g., peak oil, biofuels, the hydrogen economy, Germany's Energiewende, Amory Lovins, and Elon Musk) [1]. Smil's thoughts and assessments, however, occasionally display surprising paradoxes and contradictions. As a young man trapped on the wrong side of the Iron Curtain, he refused to join the Community Party and later studied in detail the abysmal environmental outcomes delivered by central planning [16,17]. Yet, he has long disparaged conspicuous consumption, unlimited free trade, and free-market economists [18] while sometimes sounding as though he would relish the opportunity to redesign our economies along less energy-intensive lines [19]. Intellectually closer to natural scientists who consider humans ultimately no different than other animal species, he described and illustrated in abundant detail our uniqueness through various studies in the history of technology [20]. A proponent of reducing humanity's resource footprint through smaller and more efficient homes and cars, he is critical of large-scale urbanization because it facilitates wealth creation and results in greater per capita energy consumption [11,21]. A political refugee who made a good living in Canada and a long-time member

of the academic globe-trotting elite, he could never side with prominent American environmentalists who called for immigration restrictions in the name of curbing economic growth [22], yet his policy recommendations would ultimately deny the opportunities he has enjoyed to most of humanity [11,19].

To this Promethean writer, the most surprising paradox in Professor Smil's work is his persistent Malthusianism, for it not only flies in the face of much empirical data, but it also contradicts his own earlier writings on limits to growth, Peak Oil and the history of technology as synthesized in his 2017 book "Energy and Civilization: A History" (henceforth *Energy and Civilization*) [21]. In the remainder of this essay, I argue that Smil's ideological blinders, as expressed in *Growth*, *Energy and Civilization* and elsewhere, prove too overwhelming to him. Like most of us, he ultimately can't resist the temptation to torture the evidence until it confesses his pre-ordained pessimistic conclusions, thereby inviting this more optimistic—and similarly one-sided—rebuttal.

This essay is organized in two main parts. The first sketches out some of Professor Smil's broader contributions and his neo-Malthusian outlook. This is followed by a detailed presentation of the Promethean perspective and a critical discussion of some of Smil's key assertions. In keeping with his historical outlook, I rely mostly on Promethean contributions that long predate the modern environmental movement—and can thus avoid charges of "greenwashing"—to argue that, in the right institutional context (i.e., market economies characterized by competition, the price system, and private property rights), humans' unique propensity to trade physical goods and to (re)combine existing things in new ways ultimately deliver both wealth creation and reduced environmental harm. I also challenge Professor Smil's casual dismissal of the "circular economy", at least to the extent the concept suggests it is not the volume of materials handled, but rather how they are handled, that determines the environmental impact of economic growth. Contra Smil, the way forward does not lie in a politics of less, but rather in the embrace of what makes us humans.

## 2. The Neo-Malthusian Polymath

Smil's lengthier books are remarkable for their scope and sheer amount of detailed factual information. Their narrative structures and editorial choices, however, often suffer from the fact that their author rarely seems to have a specific audience in mind. *Growth* and *Energy and Civilization*, each over 500 pages in length, are typical of his earlier output.

### 2.1. Summa Smilia

As historian of technology David E. Nye blurbed on the dust-jacket cover, *Energy and Civilization* can be considered the "definitive work on this vital subject". The book is a revised and expanded version of the earlier primer "Energy in World History" [23]. As highlighted in the preface, the new text is 60% longer, has 40% more images, more than twice as many references, and builds upon the work of hundreds of scientists, engineers, historians, and economists. Taken together, the addenda, notes and bibliography are the size of a small book. Stylistically, the text consists mostly of short detailed vignettes. Many of these describe a group of people or a specific inventor who contributed the process or product X. X displayed features Y that, in turn, increased productivity by a factor of Z. Smil also throws in a few calculations of his own on subjects such as the energy costs of Roman roads, pre-industrial wind and energy power, the amount of energy required to sustain a female coal bearer in an early nineteenth century Scottish mine, and the power densities of past urban energy supply and use.

Along the way readers learn much beyond the history of technology narrowly construed. Browsing randomly through the book, one comes across the historical use of pack camels in the arid region between Afghanistan and Morocco, the game-changing nature of hydraulic fracturing on the global supply of petroleum products, the nature and productivity of Pre-Columbian agriculture, the technical details and devastation of the firebombing of Tokyo in March 1945, the 2015 revised estimates of global energy subsidies compiled by the International Monetary Fund, and the murderous character of Stalin's and Mao's regimes. Some of the more exotic references include an 1880 English

translation of "The Lusiads" (the 1572 epic Portuguese poem describing Vasco de Gama's discovery of a sea route to India), the 1945 "United States Air Force Statistical Digest World War II", Robert Caro's massive biography of American President Lyndon B. Johnson, and articles published in outlets such as "Études françaises" and "Extreme Physiology & Medicine".

While rejecting the label of energy determinist, Smil's core message is nevertheless that "[c]ivilization's advance can be seen as a quest for higher energy use required to produce increased food harvests, to mobilize a greater output and variety of materials, to produce more, and more diverse, goods, to enable higher mobility, and to create access to a virtually unlimited amount of information" (pp. 417–418). The ultimate result has been "larger populations organized with greater social complexity into nation-states and supranational collectives, and enjoying a higher quality of life" (p. 385). Smil, however, expresses concern about "environmental externalities" and inequality between those who consume too much ("high-energy countries") and those who have too little (p. 441). Because of these, he calls for a "commitment to change" (p. 441). Unfortunately, nothing of substance will happen in the short run because energy systems are too complex to be significantly altered by subsidies to technically inferior alternatives, be they wind power or liquid biofuels.

*Growth* covers some the same ground, but it is a different animal in terms of its ambitions and ultimate goals. Smil describes the book as the most comprehensive study on the modalities of growth in Earth's life systems in their many "natural, social, and technical forms" [11] (p. xix), meaning biological organisms, human artifacts (from simple tools to complex machines) and complex anthropogenic systems (from population to economic growth) (Systems are defined as "entities consisting of connected and interdependent parts that make up specific structures and provide desired functions" (p. 303)) The first chapter introduces readers to common growth patterns (e.g., linear, exponential, hyperbolic, sigmoid) and outcomes (e.g., normal and power-law distributions). This is followed by a reworking and repackaging of some the author's earlier writings on agriculture, technological change, human societies, and other subjects. In the last chapter, Smil puts on his activist hat and worries about environmental abuses and the fate of humanity.

As expected, Smil punctuates his core text with a few quirky calculations such as the number of musical masterpieces Mozart could have produced had he lived beyond the age of thirty-five. He also compiles yet another list of detailed studies written by knowledgeable experts who proved completely clueless about (often short-term) future developments and outcomes. Failed predictions, Smil argues, can often be explained by small variations or external interventions that disrupt the neat progression of an expected growth trajectory in both simpler systems (e.g., weather events on a crop) and in human societies. A striking case of the latter is the unexpected development of hydraulic fracturing that turned the American petroleum production output curve from an asymmetrical bell shape into something of an incomplete bimodal distribution. Readers are also warned never to infer simple and predictable growth patterns on rapid initial take-offs. Cases in point include the historical trajectories of British coal extraction, the sudden halt of Japanese economic growth a generation ago, and a sharp decline in electric car sales.

Although this is obviously not the author's intent, social scientists and historians already skeptical of crude inferences from the methods and patterns of outcome observed in the domains of the natural sciences to the realm of human actions will feel rather vindicated by Smil's evidence. After all, what can growth patterns observed in geological processes or simpler life forms really tell us about creative individuals willing to dissent from the status quo, consumers with evolving preferences, or market economies characterized by an ever more complex and refined division of labor? What is the analog to rapid and sometimes drastic changes brought about by political actors, such as the displacement of laissez faire and fiscal conservatism by new tax and spend policies, targeted "green" subsidies and bans on conventional energy production and transport? Smil agrees to an extent, especially when discussing the "role of the basic sociopolitical setting on economic growth and individual prosperity" by "comparing outcomes in countries controlled for decades by Communist parties with the achievement of neighboring nations" (e.g., North and South Korea, East and West Germany) (p. 417). In the end

though, he reverts to the default position of most environmental scientists that humans should be treated like other living creatures and ecosystems because they are subjected to immutable natural laws.

## 2.2. *"Numbers Don't Lie", but They Don't Tell the Right Story, Either*

Smil's neo-Malthusian outlook permeates much of his work. His recent output is no exception. In one of *Growth*'s few paragraphs without numbers or references, he contends that Malthus' basic assumption is "unassailable" for "the power of population growth is indeed much greater than the capacity to produce adequate subsistence", at least when "population growth is unchecked" [11] (p. 317). He does acknowledge, however, that Malthus was wrong to think that a food shortage would keep populations in check. Smil further admits that economic growth proved to be the best contraceptive. As he is often fond of stating, "numbers don't lie". Strangely though, the remarkable increases in agricultural yields and total food production do not ultimately tell a tale of human achievement and creativity, but rather of humanity having tapped into its finite store of natural resources and having failed miserably to tackle environmental impacts. These are said to constitute "high direct and indirect energy subsidies" [11] (p. 317).

Unlike most economists who have long viewed an ever expanding division of labor as the necessary foundation of further growth and development, Smil, like many other sustainability theorists before him, laments instead the "loss of such valuable, flexible skills as the ability to grow one's own food or to repair a range of small machines" [11] (p. 439). Readers, however, are not told how much one should push back against the division of labor. Is a hobby gardener allowed to buy pre-packaged seeds and plastic sheeting? Can a hobby mechanic plug his equipment in a functioning electrical outlet and grid? Most importantly, how is growing one's own food a better use of one's limited time than investing in the further development of specialized marketable skills and, in the process, better provide for one's family? Indeed, how many pioneers who could build a log cabin and grow food without external inputs wouldn't want to trade their situation with that of a relatively less skilled 21st century delivery truck driver?

Smil's key conclusion is straightforward: One should not be blinded by Moore's law (i.e., the doubling approximately every two years of transistors and other components emplaced on a silicon wafer) for all natural growth on Earth, including economic growth, must eventually end. In a few paragraphs he tells his readers that economic growth is ultimately unsustainable because it is based on "anthropogenic insults to ecosystems" [11] (p. 492). Dematerialization is a mirage because "decoupling economic growth from energy and material inputs contradicts physical laws". A circular economy is nonsensical for "modern economies are based on massive linear flows". In the end, the dominant model of economic progress is incompatible with the preservation of a habitable biosphere and humanity must "put an end to material growth and forge a new society that would survive without worshiping the impossible god of continuously increasing consumption" [11] (p. 497).

While Smil mentions or refers to a few sophisticated Promethean writers such as Merrill K. Bennett, Julian Simon, Herman Kahn, and Joel Mokyr, like most sustainable development theorists he proves incapable of engaging them seriously. Instead, he dismisses pro-growth policy analysts as being disproportionately "economists, lawyers, and techno-optimists" who "rarely think about the biosphere's indispensability for the survival of human societies" [11] (p. 510). To the extent he discusses the economic way of thinking, Smil mostly limits himself to criticizing concepts or approaches that have long been the favorite whipping boys of environmental studies scholars, e.g., GDP statistics, models that treat innovation as an exogenous factor, and a failure to appreciate the fundamental role of energy in economic activities. In short, his neo-Malthusianism is of the kind one encounters at most environmentalist academic meetings, for not only is techno-optimism boiled down to a soulless straw human, but it is also denied a competent defense and drowned under incantations of Kenneth Boulding's famous mantras that "Anyone who believes in indefinite growth in anything physical, on a physically finite planet, is either mad or an economist" or else that society needs to move from a "cowboy economy" to a "spaceman economy" [11] (pp. 492–493).

To a Promethean reader, *Growth* is more akin to a grandiose Malthusian legal brief than a balanced assessment of an ancient and perennial debate. This will not surprise long-time readers of Smil, for the book doesn't contain anything that he hasn't already stated in more detail before. For instance, in his 2013 book *Made in the USA* Smil disparaged the "growth imperative of modern economies" as "obviously unsustainable" because of the second law of thermodynamics [18] (p. 11). He even went as far as rephrasing his conclusion by using the titles of then popular pessimistic books: Because "materials matter . . . we should stop shoveling fuel for a runaway train of economic growth . . . embrace the logic of sufficiency . . . confront consumption . . . and make a break with the throwaway culture . . . by reasserting self-control" [18] (pp. 11–12). Smil's thinking is also what one expects from the author of another book titled "Harvesting the Biosphere. What we have taken from nature" [24]. As he further re-stated in a recent interview: "This is a finite planet. There is a finite amount of energy. There is finite efficiency of converting it by animals and crops. And there are certain sensitivities in terms of biogeochemical cycles, which will tolerate only that much. I mean, that should be obvious to anybody who's ever taken some kind of kindergarten biology" [19] (n.p.).

Yet, Professor Smil also knows that the core arguments put forward by neo-Malthusians over the last two centuries have not withstood the test of time in market economies. As he documents in *Growth* and many other writings [25], there is now so much food in most parts of the world that many poor people suffer from obesity rather than calorie deficiencies. In *Energy and Civilization*, he specifically dismisses concerns about "the rising use of fossil fuels [as] a cause for concern about their early exhaustion" or "the early onset of unbearably rising real costs of recovering these resources" [21] (pp. 424–425). Indeed, not only has there never a shortage of non-renewable commodities in functioning market economies, but the inflation-adjusted price of most valuable commodities has often gone down drastically, indicating greater availability in spite of rapidly rising consumption [26].

Faced with the facts that supply numbers don't lie, many neo-Malthusians have long pivoted towards claims of present or future irreversible environmental degradation as a result of population growth, overproduction, and overconsumption [27]. As perhaps best stated over five decades ago by Resources for the Future staffer Henry Jarrett, the "underlying causes" of problems such as "lowered environmental quality, smoggy atmosphere, polluted streams, noise [and] land skinned by strip mining" have long been believed by many "to be seen in the same statistics that most of the time are hailed as indicators of economic growth" [28] (pp. viii–ix). Smil shares this assumption and consequently calls for learning to live within solar and biospheric limits, which would involve a "delinking of social status from material consumption" [21] (p. 440). Yet, in a harsh review of Jared Diamond's best-seller "Collapse", Smil also commented that the most frequently invoked historical examples of environmental destruction as a result of population growth and alleged unsustainable societal practices (e.g., Rapa Nui, Norse settlements in Greenland) failed to conform to the environmentalist narrative and were at any rate of little if any relevance for more resilient advanced societies [29]. Smil's writings on the Chinese environment also suggest that economic growth coupled with even partial institutional reform can go a long way in addressing seemingly hopeless predicaments [30].

Like many present-day environmentalists, Smil's claims of massive environmental degradation are ultimately based on rather extreme and controversial scenarios and analytical frameworks such as the sixth mass extinction [31] or the Stockholm's Resilience Center's planetary boundaries [32]. Needless to say, such catastrophist claims have a long history. For example, over a century ago scientific management guru Frederick Winslow Taylor summed up contemporary fears as follows: "We can see our forests vanishing, our water-powers going to waste, our soil being carried by floods into the sea; and the end of our coal and our iron is in sight" [33] (p. 5). The President of the New York Zoological Society and prominent eugenicist Henry Fairfield Osborn similarly observed at the time that, with the exception of conservation areas, nowhere was "nature being destroyed so rapidly as in the United States" [34] (n.p.). Not only did "vulgar advertisements hide the landscape", but "air and water are polluted, rivers and streams serve as sewers and dumping grounds, forests are swept away, and fishes are driven from the streams. Many birds are becoming extinct, and certain mammals

are on the verge of extermination" [34] (n.p.). While there were indeed severe problems at the time, they proved largely manageable in later decades. Indeed, environmental indicators such as air and water quality and the extent of the forest cover suggest that the American environment is now in a better (or at least less bad) state than it was a century ago despite a population that more than tripled in size and is now considerably wealthier [35]. In the end, if not all environmental trends are good everywhere, one can hardly deny that, with the exception of carbon dioxide emissions, most environmental indicators in societies that have experienced significant economic growth have long shown (often major) improvement over time [35,36].

Smil also echoes another long-standing complaint of neo-Malthusians, i.e., the deleterious impact of greater population numbers and increased material wealth on the need for the psychological (or spiritual or affective) flourishing one experiences as a result of encountering natural beauty, avoiding large crowds and enjoying other life forms ("biophilia"). In his case, this takes the form of light pollution that interferes with his enjoyment of a "starry sky bisected by the Milky Way" or East Asian tourists who ruin his contemplation of Velázquez's "Las Meninas" at the Museo del Prado [11] (p. 499). In this, Smil echoes illustrious predecessors such as economist John Stuart Mill who commented in 1848 that if there was indeed enough room for population growth "supposing the arts of life to go on improving" and "capital to increase", it was nonetheless "not good for man to be kept perforce at all times in the presence of his species" because a "world from which solitude is extirpated is a very poor ideal" [37] (n.p.). Solitude, he argued, was essential to improve one's character, such as when meditating in "the presence of natural beauty and grandeur".

A century later, best-selling eco-catastrophist author William Vogt wrote that contemplating the "Peruvian Andes, high above the timberline, where the vast and ancient movement of the earth's crust lies recorded before the eyes of any observer who will stop to look, are so majestic, so awe-inspiring", that it created in him an experience similar to listening to Beethoven's Ninth Symphony [38] (pp. 94–95). To people able to appreciate the perfection and richness of natural beauty, Vogt added, environmental destruction arose the same kind of feelings that a Frenchman would experience if someone slashed the Mona Lisa. In his 1957 presidential address to the Population Association of America, prominent population economist Joseph Spengler stated that "an overworked stork is the enemy of the beautiful" [38] (p. 61). His 1965 presidential address to the American Economic Association similarly decried Americans "prepared to trade natural grandeur and 'spontaneous activity of nature'" for "junkyards and carscapes" in a failed attempt to access "'God's great open spaces'" [39] (p. 5). Indeed, Spengler observed, "some hold [economist] J. K. Galbraith had better label ours an effluent society than an affluent one" [39] (p. 10).

While environmental regulations and public education campaigns undoubtedly played a role in cleaning up the environment of ever wealthier societies, what Smil and other prominent neo-Malthusians have long failed to grasp is that market-based economic development has always contained the seeds of both improved standards of living and environmental remediation. I now turn to a discussion of their key arguments.

## 3. Homo Prometheanus

In the last 70,000 years *Homo sapiens*' numbers grew from thousands of hunter-gatherers to nearly eight billion individuals, a majority of whom are now city-dwellers. The outcomes of this process in terms of life expectancy, health and many other indicators have been rather spectacular [35]. Needless to say, progress would have been unthinkable if some of our ancestors had not challenged existing ways of doing things and invented new things to do. As the economist Erich Zimmermann observed nearly a century ago, before the emergence of humans "the earth was replete with fertile soil, with trees and edible fruits, with rivers and waterfalls, with coal beds, oil pools, and mineral deposits; the forces of gravitation, of electro-magnetism, of radio-activity were there; the sun set forth his life-bringing rays, gathered the clouds, raised the winds" [40] (p. 3). However, he added, "there were no resources". Resources, he later explained, were in reality "highly dynamic functional concepts; they are not,

they become, they evolve out of the triune interaction of nature, man, and culture, in which nature sets outer limits, but man and culture are largely responsible for the portion of physical totality that is made available for human use . . . knowledge is truly the mother of all resources" [41] (pp. 814–815).

Prometheans have long traced resource creation back to some unique human attributes. The first is, in Adam Smith's famous words, the "propensity to truck, barter, and exchange one thing for another" which has long been "common to all men, and to be found in no other race of animals" [42] (n.p.) Whether it caused or was a result of it, individuals came to specialize ever more narrowly in what they did best and, trading with each other, produced far more, both in terms of quantity and quality, than if each individual or larger social group had remained self-sufficient.

Another key characteristic is the capacity to solve problems of all kinds by constantly (re)combining existing things in new ways. It was thus common a few decades ago to distinguish between the 'retardation school' of technological change, whose proponents believe that "the more that has been invented the less there is left to be invented", and the 'acceleration school' according to which "the more that is invented the easier it becomes to invent still more" because "every new invention furnishes a new idea for potential combination with vast numbers of existing ideas" and the "number of possible combinations increases geometrically with the number of elements at hand" [43] (p. 156). Nearly a century ago, the radical American historian Charles Beard thus observed that there can never be anything final about technological advances for the "solution of one problem in technology nearly always opens up new problems for exploration" and "[a]ctivities in one specialty produce issues for its scientific neighbors" [44] (p. xxiv). Beard saw no end to this process because of the "passionate quest of mankind for physical comfort, security, health, and well-being". He added that until "people prefer hunger rather than plenty, disease rather than health, technology will continue to be dynamic" and that "[c]uriosity would have to die out in human nature before technology could become stagnant, stopping the progress of science and industry".

Writing from the perspective of the inventor and technician, the engineer and historian of technology Henry Petroski argued that the "form of made things is always subject to change in response to their real or perceived shortcomings, their failures to function properly. This principle governs all invention, innovation, and ingenuity; it is what drives all inventors, innovators, and engineers" [45] (p. 22). Furthermore, "since nothing is perfect, and, indeed, since even our ideas of perfection are not static, everything is subject to change over time. There can be no such thing as a 'perfected' artifact; the future perfect can only be a tense, not a thing". In other words, successful market innovations mandate the creation of smaller or less important problems than those that existed previously. As the Canadian engineer and communist activist Herbert Dyson Carter observed over eight decades ago, commercially successful inventions must either save time, lower costs, last longer, do more, work better or sell more easily [46] (p. 143). While not all of these characteristics have environmental benefits, most do.

In what follows I flesh out in more detail the Promethean perspective and challenge some of Smil's assertions by calling upon authors whose writings long predate the modern environmental movement. This will illustrate both how long-standing and vindicated Promethean arguments have been over time.

### 3.1. Human Exceptionalism

In his "Treatise on Political Economy" first published in 1803, French liberal economist Jean-Baptiste Say was confident that humans' unique abilities to trade, barter, reason and use foresight would give them the ability to offset natural constraints, unlike other animals that were "incapable of providing for future exigencies" [47] (n.p.). As such, people were "not more scantily supplied with the necessaries of life, because their number is on the increase", nor more materially prosperous "because it is on the decline". Rather, their "relative condition depends on the relative quantity of products they have at their disposal; and it is easy to conceive these products to be considerable, though the population be dense; and scanty, though the population be thinly spread". Indeed, he observed, famine was more

common in Europe during the less populated Middle Ages than in his days. In one of his "Letters to Malthus", Say explicitly dismissed the belief that a reduction in population would "enable those which are left to enjoy a greater quantity of those commodities of which they are in want" because it failed to grasp that a reduction in manpower simultaneously destroyed the means of production [48] (n.p.). After all, one did not see in thinly populated countries that "the wants of the inhabitants are more easily satisfied". To the contrary, it was "abundance of productions, and not the scarcity of consumers, which procures a plentiful supply of whatever our necessities require".

Anarchist theorist William Godwin similarly observed at the time that a human being is the "only animal capable of persevering and premeditated industry" and the "only creature susceptible of science and invention, and possessing the power of handing down his thoughts" [49] (n.p.). As a result of past advances, most humans were "not living upon the wild fruits of the earth, or the wild animals of the field", but upon the products of human industry. Every person born into this world was therefore "a new instrument for producing the means of subsistence" and every member added to the numbers of the community, is a new instrument for increasing those means". The human species was therefore "capable of improvement from age to age, by means of which capacity we have arrived at those refinements of mechanical production and science, which have been gradually called into existence". By contrast, "all other animals remain what they were at first, and the young of no species becomes better or more powerful by the experience of those that went before him".

Building on Say's work, in the middle of the nineteenth century the French liberal economist Frédéric Bastiat granted Malthus his key premise for "all living species, except man" [50] (n.p.). A human being, he wrote, "is perfectible; he seeks to improve his situation" and "[p]rogress is his normal state". As a result, "the means of existence increase more rapidly than population" (italics in original). Bastiat argued his case "not only [based on] the theory of perfectibility", but also on "facts, since everywhere we find the range of man's satisfactions widening" (italics in original). Karl Marx later wrote that an "abstract law of population exists for plants and animals only, and only in so far as man has not interfered with them" [51] (n.p.).

In his 1879 best-selling "Progress and Poverty", American economist Henry George stated that "everywhere the vice and misery attributed to over-population can be traced to the warfare, tyranny, and oppression which prevent knowledge from being utilized and deny the security essential to production" [52] (n.p.). This was because, of "all living things, man is the only one who can give play to the reproductive forces, more powerful than his own, which supply him with food". Indeed, "while all through the vegetable and animal kingdoms the limit of subsistence is independent of the thing subsisted, with man the limit of subsistence is, within the final limits of earth, air, water, and sunshine, dependent upon man himself". As he put it, if "bears instead of men had been shipped from Europe to the North American continent, there would now be no more bears than in the time of Columbus, and possibly fewer, for bear food would not have been increased nor the conditions of bear life extended, by the bear immigration, but probably the reverse". It was therefore "not the increase of food that has caused [the North American] increase of men; but the increase of men that has brought about the increase of food".

In a book first published in 1957, the Austrian economist Ludwig von Mises observed that only humanity had the power to escape from the struggles for survival, provided people engage in social cooperation within the context of a market economy. As he saw things, an "eminently human common interest, the preservation and intensification of social bonds, is substituted for pitiless biological competition, the significant mark of animal and plant life" [53] (p. 56). As a result, humanity was "no longer forced by the inevitable laws of nature to look upon all other specimens of his animal species as deadly foes". It was thus "inappropriate to refer to animals and plants in dealing with the social problems of man" because for "animals the generation of every new member of the species means the appearance of a new rival in the struggle for life. For man … it means rather an improvement than a deterioration in his quest for material well-being".

The American Trotskyist Joseph Hansen also noted three years later that he and his comrades took "a decidedly different view of humanity" than neo-Malthusians because they "note that man has hands and a brain, the capacity to use tools and an inclination for teamwork. These have made him, in distinction to all other animals, a food producer" [54] (p. 43). Hansen added that, in "today's world, hunger is completely abnormal. Humanity can produce all it needs and many times over. Moreover, man's capacity to increase his food supply expands with the increase in population and at an ever-higher rate than population growth". A big population was therefore "an asset, not a liability. Failure to see this rather obvious fact is the basic flaw in the Malthusian argument".

### 3.2. Trade, Innovation, and Resource Creation

Prometheans have long argued that a larger population lays the foundation for greater resource creation. As British political economist William Petty observed over a century before Malthus, it was "more likely that one ingenious curious man may rather be found out amongst 4,000,000 than 400 persons" [55] (p. 49). The economist Edward Cannan objected a century ago to the notion that agricultural productivity would have been greater in his days if population numbers had remained small because its proponents failed to understand that fewer brains meant that fewer advances would "have been discovered and introduced" [56] (n.p.). More recently, the physicist Robert Zubrin asked who between Louis Pasteur and Thomas Edison should not have been born in order to improve the lot of humankind [57] (p. 24).

Needless to say, sheer numbers by themselves would mean little in the absence of an ever broader and integrated division of labor and the technological advances it makes possible. In a personal reply to and further face-to-face conversation with Malthus, the American diplomat Alexander Everett suggested that an expanded division of labor not only made people more productive, but further laid the foundation for "the invention of new machines, an improvement of methods in all the departments of industry, and a rapid progress in the various branches of art and science" that resulted in a level of labor productivity that far exceeded the proportional increase in population numbers [58] (p. 26). A belief in decreasing returns, he argued, ultimately assumed that "labor becomes less efficient and productive in proportion to the degree of skill with which it is applied; that a man can raise more weight by hand, than by the help of a lever, and see further with the naked eye than with the best telescope" (p. 28). In 1844, Friedrich Engels stood Malthus on his head by observing that "science increases at least as much as population. The latter increases in proportion to the size of the previous generation, science advances in proportion to the knowledge bequeathed to it by the previous generation, and thus under the most ordinary conditions also in a geometrical progression" [59] (n.p.).

Henry George noted three decades later that while one could see "many communities still increasing in population", they were also "increasing their wealth still faster" [52] (n.p.) Indeed, "among communities of similar people in a similar stage of civilization", the "most densely populated community is also the richest" and the evidence was overwhelming that "wealth is greatest where population is densest; that the production of wealth to a given amount of labor increases as population increases. These things are apparent wherever we turn our eyes". In the end, the "richest countries are not those where nature is most prolific; but those where labor is most efficient—not Mexico, but Massachusetts; not Brazil, but England". Where nature provides modest resources, George commented, "[t]wenty men working together will . . . produce more than twenty times the wealth that one man can produce where nature is most bountiful". This was because the "denser the population the more minute becomes the subdivision of labor, the greater the economies of production and distribution, and, hence, the very reverse of the Malthusian doctrine is true; and, within the limits in which we have reason to suppose increase would still go on, in any given state of civilization a greater number of people can produce a larger proportionate amount of wealth, and more fully supply their wants, than can a smaller number".

A concise overview of the anti-Malthusian stance was printed in 1889 in the Westminster Review:

The Malthusian theory does not accord with facts. As population grows, instead of production being less per head, statistics clearly prove it to be greater. The intelligence which is fostered in large communities; the advantages of the division of labour; the improved transit, which increases in efficiency with an enterprising people in proportion as numbers become large, and is impracticable until population has developed—are more than a match in the competition of production for any advantage a thinly scattered community may in some respects gain on a virgin soil. Malthus and his followers, while bringing prominently forward the needs of an increasing population, keep out of view the increasing means of supply which the additional labour of greater numbers will produce . . . . and so long as there are a pair of hands to provide for every mouth, with intelligence and energy ample production is assured, unless society erects artificial barriers by means of its laws regarding the distribution of wealth. [60] (p. 287)

In his 1944 "The Theory of Economic Progress", economist Clarence Ayres emphasized the importance of "the principle of combination" to human creativity and applied it in a variety of different ways. The exponential growth or proliferation of technical devices could thus be explained because "the more devices there are, the greater is the number of potential combinations" and because the cross-fertilization of ideas was a key component of the discovery process [61] (n.p.). In this context, the supply of natural resources could never be static:

The history of every material is the same. It is one of novel combination of existing devices and materials in such fashion as to constitute a new device or a new material or both. This is what it means to say that natural resources are defined by the prevailing technology, a practice which is now becoming quite general among economists to the further confusion of old ways of thinking (since it involves a complete revision of the concept of "scarcity" which must now be regarded as also defined by technology and not by "nature". [61] (n.p.)

The result of these processes, as had been understood in embryonic form nearly three centuries earlier by the German alchemist Johann Joachim Becher, was that with "increase of population come increased facilities for subsistence, and through the latter comes influx of people; this in its turn causes further increase of population, and so on in an everlasting circle" [62] (p. 154). Writing in 1771, the French economist Nicholas Baudeau similarly argued that the "productiveness of nature and the industriousness of man are without known limits" because production "can increase indefinitely" and as a result "population numbers and well-being can go on advancing together" [63,64] (p. 98).

In the end, Prometheans, whatever their political leanings, typically shared four core beliefs: (1) humans differ sufficiently from other animal species to invalidate analogies between growth in human societies and growth in other social animals and ecosystems; (2) innovation is cumulative; (3) an ever greater division of labor and a growing population lay the foundations for future growth and wealth creation; (4) bad economic and environmental outcomes should be blamed on poor governance rather than population growth. Prometheans, however, disagreed as to the practical policy implications of this last belief based on their political affinities.

### 3.3. Incentives, Institutions, and Green Innovations

While Professor Smil obviously understands the power of economic incentives, he frequently expresses doubts about the outcome of free markets. In *Growth* he laments the cramped living conditions of broilers (i.e., chickens raised for meat) and pigs, but understands that farmers cannot be profitable with lower densities [11] (p. 148). Another problem that seemingly rattles him a great deal is that houses, appliances, personal vehicles, and many other devices keep getting more numerous, bigger, and more complex over time [11] (pp. 8–9). In a recent interview, Smil discussed the small super-insulated house he built in Winnipeg (Canada), an urban jurisdiction that has long had some of the cheapest land, real estate, and energy costs (especially hydro-electricity rates) in the developed world. As he admits, his creation made no sense economically and people don't want to pay a premium

for it "unless the price of energy goes up, up" [19] (n.p.). (Luckily for him, years of bad governance at Manitoba Hydro will soon make higher energy prices a reality in his province [65].) If everybody had such a house, he argues, humans would emit between five and eight billion fewer tons of carbon dioxide every year, an outcome that proves the existence of "enormous slack in the system" [19]. Following this argument to its logical conclusion, however, one might as well consider children the worst form of "slack" to be gotten rid of, a position now promoted by a seemingly growing number of climate change activists [66].

To someone like Smil who views human societies through neo-Malthusian lenses, the price system is obviously inadequate. To others, however, prices remain the best way to factor in innumerable trade-offs in order to achieve a rational (i.e., economic) allocation of scarce resources out of an incredibly large number of possible combinations. Market outcomes are therefore not determined by ruthless displays of corporate power, but by a never-ending discovery process driven by consumer demand in which existing and new technologies and materials are continuously pitted against each other. As will now be argued, to work properly and deliver beneficial environmental outcomes, the price system must be embedded in private property rights.

### 3.3.1. Private Property Rights

To their supporters, secured private property rights have always incentivized forward thinking, innovation, greater productivity, and careful stewardship of natural resources. As John Locke argued in 1690, the "provisions serving to the support of human life, produced by one acre of inclosed and cultivated land, are (to speak much within compass) ten times more than those which are yielded by an acre of land of an equal richness lying waste in common" [67] (n.p.). A century later English agricultural writer Arthur Young famously wrote that the "magic of property turns sand into gold" and "Give a man the secure possession of a bleak rock, and he will turn it into a garden; give him a nine years' lease of a garden, and he will convert it into a desert" [68] (n.p.). Despite Malthusian inclinations, French philosopher Antoine Destutt de Tracy observed at about the same time that Lombardy and Belgium, although often ravaged by war, were "always flourishing" because of well-established private property rights. On the other hand, Poland had a small and stationary population "because its inhabitants being serfs, and wretched, have in the midst of abundance very slender means of existence" [69] (n.p.). However, he argued, "suppose for a moment the small number of men, to whom these serfs belong, and who devour their substance, driven from the country, and the land become the property of those who cultivate it, you would see them quickly become industrious, and multiply rapidly" [69] (n.p.).

Less well understood is that a system of private property rights also submits owners to legal sanctions based on constraints such as the common law (tort) doctrines of trespass (any entry on the property of another) and nuisance (indirect or intangible invasions, such as odors and noises, or any unreasonable interference with another's use or enjoyment of his property) or their equivalent in other legal traditions (e.g., civil law systems). In this context, polluting someone else's property is no more acceptable than vandalizing it. Such actions can result in damages being awarded or even an injunction (i.e., an order requiring the cessation of an offensive activity or specifying corrective action), although in practice judges historically often granted a stay of injunction, i.e., a suspension of the injunction that would allow the defendant time to work out the most convenient mode of compliance [70,71].

In practice, while such common law procedures sometimes bankrupted polluting businesses, they also often triggered creative thoughts that culminated in the development of lucrative by-products [72,73]. For instance, the journalist Peter Lund Simmonds observed a century and a half ago that the stench resulting from the blood and offal at a large pork-packing establishment "had become such an offense to the neighbourhood, that the proprietors were threatened with a perpetual injunction" [74] (pp. 39–40). Shortly afterwards, however, they developed a method through which they dried the entire refuse, including the blood. The parts containing sufficient fat to make the operation economical were first treated in a rendering tank where the clean fat was converted into lard

and the refuse into grease and grease oil. The scrap left in the process, consisting of the bones of the head and feet and considerable meat, was then thoroughly mixed with the blood, dried, and converted into a valuable output. The whole process resulted in a smell that was comparable with that of a pot of boiled cabbage. Writing almost a century ago, Erich Zimmermann similarly observed that not all businesses were "free to strive from the maximization of profit without social interference" and that "waste elimination may be enforced by law even if it does not pay in the economic sense" [40] (p. 768). It sometimes happened, however, that "a corporation compelled by legal action to eliminate a waste at great expense, and unable to pass the cost on to the consuming public, may succeed, with the aid of scientific research, in converting the waste products into paying by-products—perhaps, even into a product of major importance".

### 3.3.2. Market Prices

The short version of the case for the price system is as follows. Resources are always scarce, while human needs and desires are not. The interaction of supply and demand results in prices that reflect the relative scarcity of physical and intellectual resources. Profits and losses are generated by individuals' relative ability to combine scarce inputs in order to provide products and services that consumers value more than the available alternatives. Over time, goods more valuable than the sums of the inputs taken separately get produced, while goods worth less than the sum of their inputs are not. The appropriate measure of a firm's success in creating value is therefore long-term profitability.

Less appreciated is that market prices have long promoted both resource creation and greater efficiency in resource use. When the price of a commodity increases, market actors look for more of it, use it more efficiently and develop substitutes. As a result, resources for which there is a sustained demand have become more abundant while their inflation-adjusted prices have decreased [26]. Even in the context of stagnant or declining prices, farmers, manufacturers, and others have to keep up with their competitors and become ever more creative and efficient over time, an outcome documented by Smil himself in several books.

### 3.3.3. Underground Resources

As Smil knows better than perhaps anybody else, much resource creation in the last two centuries has resulted in the substitution of resources produced or harvested from biomass on the surface of the planet by substances ultimately extracted from below the ground and transported over ever longer distances. Carbon fuels, metals, sand, clay, silicon, potash, and phosphate, among others, have thus allowed, through many transactions, much developmental work and constantly improved manufacturing processes, to drastically reduce overall demand for wild fauna such as whales (e.g., whale oil, baleen, perfume base), birds (e.g., feathers), elephants, polar bears, alligators and countless other wild animals (e.g., ivory, fur, skin); trees and other plants (e.g., lumber, firewood, charcoal, rubber, pulp, dyes, green manure); agricultural products (e.g., fats and fibers from livestock and crops, leather, dyes and pesticides from plants); work animals (e.g., horses, mules, oxen); and human labor in various forms (e.g., lumbering, weeding). Smil is also aware that, in economies that benefitted from drastic productivity increases and wealth creation that owed much to underground resources, much marginal agricultural land has been abandoned and allowed to spontaneously revert back to a wild(er) state. Somehow though, he seems unable to conceive how vigorous economic growth can spontaneously occur with a simultaneous greening of our planet.

What Smil apparently can't see has long been obvious to a few Promethean analysts. Writing in 1933, Erich Zimmermann observed that as "science becomes bolder and more efficient", the "movement away from nature gains momentum and, in extreme cases, production rests only indirectly on "land" and is freed from the limitations which a direct dependence on "land" involves" [40] (p. 762). In the past, he wrote, "nature was the only reservoir from which man drew the raw materials of production", but in his time industry had not only learned to tap into the waste heap and the junk pile (e.g., scrap steel), but also to create "artificial substitutes, especially synthetic products". This had

come about "[w]henever the price of a natural product, essential to some industry, [rose] as the result of the absolute limitation of the actual supply or of monopolistic control over the supply" As a result, "chemists throughout the industrial world strain[ed] every effort to imitate the product or to find usable substitutes. A high market price tends to stimulate research" (p. 764).

During the Second World War, the geologist Kirtley Fletcher Mather noted approvingly that a "hundred years ago, nearly 80 per cent of all the things men used were derived from the plant and animal kingdoms, with only about 20 per cent from the mineral kingdom. Today only about 30 per cent of the things used in industrialized countries come from things that grow; about 70 per cent have their sources in mines and quarries" [75] (p. 55).

As the agricultural economist Karl Brandt also observed at the time:

> During World War I and immediately after, the belief was common among scholars and statesmen that Malthus' doctrine was still valid and that, owing to the progressive propagation of man, scarcity of food was not only inevitable in the long run but characteristic also for the second quarter of the twentieth century. A few years after the war the situation in world market contradicted those assumptions. The war had fostered rapid progress in farm technology. It brought the internal combustion engine into general use for agriculture, first in America and later elsewhere. The truck, tractor, and combine were some of the machines in which it was applied. Millions of horses were replaced, and millions of feed acres were released for food production. Enormous savings in manpower and in production costs became possible. New varieties of plants made available for crop production many areas that previously could be used only for scanty grazing. Research in animal nutrition and genetics also led to much greater efficiency in converting feed into animal products. The really revolutionary progress in food production technology revealed the economic fallacy of the more than century-old secular "law of diminishing returns", as commonly applied to food production. It became apparent that technological progress made increasing economic returns and a lowering of the costs of food production possible within sufficiently wide boundaries. [76] (pp. 135–136)

Looking at a broader range of activities, the historical demographer and economic historian Edward Anthony Wrigley later argued that the organic economy

> escaped from the problem of the fixed supply of land and of its organic products by using mineral raw materials. Thus, the typical industries of the [Industrial Revolution] produced iron, pottery, bricks, glass and inorganic chemicals, or secondary products made from such materials, above all an immense profusion of machines, tools and consumer products fashioned out of iron and steel. The expansion of such industries could continue to any scale without causing significant pressure on the land, whereas the major industries of an organic economy, textiles, leather and construction, for example, could only grow if more wool, hides or wood were produced which in turn implied the commitment of larger and larger acreages to such ends, and entailed fiercer and fiercer competition for a factor of production whose supply could not be increased. Meeting all basic human needs, for food, clothing, housing, and fuel, inevitably meant mounting pressure on the same scarce resource. [77] (p. 5)

### 3.3.4. Decoupling

In sustainability parlance, relative decoupling refers to environmental impacts growing at a slower rate than population or consumption. This is achieved through productivity gains, from increased agricultural yields to lower energy inputs per unit of output. Absolute decoupling describes declining overall impacts, independent of population and consumption trends. It is most commonly achieved through resource substitution such as the reduction in the number of work horses and mules brought about by the advent of the truck, tractor and the automobile; the reduction in greenhouse gas emissions



that resulted from the substitution of coal by natural gas in electricity generation; or the replacement of paper by electronic devices.

Smil, however, has no time for the most optimistic scenarios, for "[d]ecoupling economic growth from energy and material inputs contradicts physical laws: basic needs for food, shelter, education, and employment for the additional billions of people to be added by 2100 will alone demand substantial energy flows and material inputs" [11] (p. 492). Yes, he admits, inputs will exhibit "lower relative intensities (energy/mass, mass/mass) than today's average rates", but absolute material use will either rise with continued population and economic growth, or else "will moderate but remain substantial". Why this is inherently a bad thing, however, is left unsaid. As Smil knows very well because of his intimate familiarity with and more formal studies of the disastrous environmental outcomes created by communist central planning [16], there is simply no correlation between overall material use and environmental degradation. In other words, material poverty is not inherently environmentally virtuous, nor his greater material use necessarily bad for the environment.

As argued three decades ago by economist and demographer Mikhael Bernstam (who relied in part on Smil's [16] work), throughout the second half of the twentieth century market economies became wealthier and cleaner while centrally planned ones stagnated or even regressed while becoming increasingly polluted. Bernstam considered this outcome the "most important reversal in economic and environmental history since the Industrial Revolution" [78] (p. 334). A short version of his analysis is that this result can be attributed to the cost minimization paradigm of market economies as opposed to the input maximization of centrally planned ones. His most relevant insight for this essay, however, is that discharges into the environment declined in market economies for reasons ranging from spontaneous energy transitions (e.g., the substitution of coal and fuel oil by natural gas and hydro-electricity) to the development and adoption of better pollution control and disposal technologies made possible by increased wealth (e.g., from sewage treatment and landfilling to waste incineration and (deep) underground injection of hazardous wastes). The elimination or proper handling of waste (i.e., uselessly processed resources and economically useless production—scrap, spills, slag, discards, refuses and other processing losses; destroyed primary resources; losses of intermediary and final output in transportation and storage), rather than greater material use as a result of increased production and consumption, thus ultimately determined the impact of economic growth on the environment. Bernstam, however, didn't discuss the spontaneous and widespread creation of valuable by-products out of polluting production residuals, a widespread and market-driven outcome that long predates recent discussions of the creation of a "circular economy".

### 3.3.5. Loop Closing

Smil considers any suggestion of imminent implementation of the circular economy as "seriously misleading" as modern economies are based on "massive linear flows of energy, fertilizers, other agrochemicals, and water" [11] (p. 492). Readers, however, are not told why these flows are or will always prove inherently problematic, especially if institutional reforms (e.g., an end to subsidies that promote wasteful behavior; more stringent enforcement of private property rights) and more innovative ways to handle them are developed and implemented.

A misconception that Smil and proponents of the circular economy seem to share, however, is that the concept marks a break with past industrial practices in market economies. Yet, in the 1875 catalogue he wrote for his exhibition on the creation of valuable by-products out of production residuals, Peter Lund Simmonds commented that the "manufacturer, of course, only considers as Waste the residues of the used raw and subsidiary substances which remain on his hands after he has obtained the principal and secondary products, and these have often in his eyes little or no comparative value. Many useful bye-products and valuable industries, however, sprung out of the profitable utilization of these" [79] (p. 2). In 1886 an encyclopedia entry described how "in the earlier days" of many manufacturing branches "certain portions of the materials used have been cast aside as 'waste,'" but over time "first in one branch and then in another, this 'waste' material has been experimented

upon with a view to finding some profitable use for it; and in most instances the experiments have had a more or less satisfactory results" [80] (p. 464). A scientific retrospective published the following year highlighted "the utilization of waste materials and by-products" as a "leading feature of the Victorian epoch" [81] (p. 299). Writing in 1904, the American industrial chemist Leebert Lloyd Lamborn observed: "If there is one aspect more than any other that characterizes modern commercial and industrial development . . . it is the utilization of substances which in a primitive stage of development of any industry were looked upon as worthless" [82] (p. 16).

While legal constraints sometimes triggered creative solutions, competitive pressures were typically deemed more significant. For instance, the Scottish chemist and politician Lyon Playfair argued that "as competition becomes keen, these waste products may become the largest source of profit" [83] (p. 269). Peter Lund Simmonds similarly wrote that "one of the characteristic and salient points of modern enterprise [is] not only to allow nothing to be wasted, but to recover and utilise with profit the residues from former working" [74] (p. 4). Perhaps his most candid passage on this topic is the following:

> As competition becomes sharper, manufacturers have to look more closely to those items which may make the slight difference between profit and loss, and convert useless products into those possessed of commercial value, which is the most apt illustration of Franklin's motto that "a penny saved is twopence earned". [79] (p. 4)

In 1927 Rudolf Alexander Clemen, then the leading economic analyst of the American meatpacking industry, viewed "the development of by-products in industry [as] one of the most outstanding phenomena in our economic life" [84] (p. vii). He credited the fear of being overwhelmed by competitors in the same or other industrial sectors as the driving force in this respect. Modern conditions, he argued, made it "almost impossible materially to cut production and distribution of expense for the majority of commodities". In this context, "one of the most important opportunities for gaining competitive advantage, or even for enabling an industry or individual business to maintain its position in this new competition", was to reduce manufacturing expenses "by creating new credits for products previously unmarketable".

Contrary to Professor Smil's environmentalist dislike of large cities, many past authors alluded to the importance of agglomeration economies for successful industrial resource recovery [85,86]. Simmonds summarized the British experience by observing that larger factories were at an advantage "in consequence of the larger quantity of residues at their command, and which necessitate special machinery for working up or utilizing", but he also pointed out that "in great industrial centres, too, the waste products of a large number of works may be easily collected" [74] (p. 4). At the turn of the twentieth century the political economist Charles Devas explained the concentration of industries by, among other factors, the "greater growth of subsidiary industries, such namely as supply materials and utilize refuse, to do which for a single factory would not be worthwhile" [87] (p. 98). The journalist and author Frederick Talbot wrote in 1920 that, in order to be successful, "co-operative and individual methods [of resource recovery] can only be conducted upon the requisite scale in the very largest cities where the volume of material to be handled is relatively heavy" [88] (p. 303). This was because "waste must be forthcoming in a steady stream of uniform volume to justify its exploitation, and the fashioning and maintenance of these streams is the supreme difficulty".

Interestingly, something akin to the creation of a circular economy was attempted in some centrally planned economies in the post-World War II era. These initiatives revolved around an elaborate hierarchical input and output quota system of waste registration, collection, distribution, and reuse and a number of mobilization campaigns. As with everything else in such an economic system, however, the experiment failed as it suffered from a number of shortcomings, mainly: (1) the lack of incentives by individuals to invest time and effort into creating and producing goods that other people are willing to pay for; (2) the difficulty of allocating resources rationally in the absence of a price mechanism or when prices have been distorted by government policies; (3) the inability of a centrally planned system

to take advantage of the unique tacit knowledge and information that individuals possess about their immediate surroundings and particular line of work [89].

## 4. Conclusions

In his recent intellectual history of left-wing Prometheanism, geographer William B. Meyer observed that "exponents of green thought have been studied much more carefully (and sympathetically) than have prophets of human mastery over the earth" [3] (p. 15). This is arguably even truer in the case of free-market Protheans. Be that as it may, to many environmental scientists and sustainable development theorists, Prometheanism, whichever side of the political spectrum it might come from, implies an unacceptable belief in "human supremacy" [90]. True, humans are perhaps less unique than what was once believed to have been the case. After all, we now appreciate better than our ancestors did that many other life forms use sophisticated means of communication, transmit tacit knowledge to their offspring, use opportunity tools (e.g., sticks, stones, sponges, thorns) and can modify long-standing practices to take advantage of new opportunities [91]. We also know that some insects (e.g., leafcutter ants, some species of termites, ambrosia beetles) developed agricultural practices tens of millions of years ago [92].

Yet, one can hardly deny that modern humans have developed unique abilities such as their capacity to trade physical goods over long distances and to expand and improve upon their stock of knowledge and capital by transmitting and (re)combining ideas. The beneficial outcomes created by these traits were arguably better understood and more widely celebrated in the past [3]. For instance, writing two centuries ago, William Godwin observed that before the publication of Thomas Robert Malthus' "Essay on the Principle of Population", most people believed that an increase in population would deliver better days. Godwin saw "something exhilarating and cheerful" in this earlier spirit when humanity believed it could summon "the unlimited power we possess to remedy our evils, and better our condition" [49] (n.p.) Humans, he observed, felt they "belonged to a world worth living in".

Fortunately, a belief in progress dominated much of the second half of the nineteenth century [3], a time period (1867–1914) Professor Smil deems the most significant watershed in human history since the emergence of settled agriculture [20]. To give but one illustration, the anonymous writer of a review essay published in The Economist in 1854 argued that Malthusians beliefs were often held with a "fervour quite religious" by some of the "leading minds of society" [93] (p. 1269). Natural constraints, these people believed, ultimately mandated "no remedy [other than] starving out the people, horrible as it is". Humanity therefore had to turn its back on "great discoveries and improvements, which render humanity more productive", for they would only make things worse down the road. Fortunately, The Economist contributor wrote, Malthusian notions that the barrier to progress was "becoming more formidable" and that "progress is always in a diminishing ratio" were "flatly and emphatically contradicted by the history of society all over the world". Indeed, "as men have been multiplied [so much faster than formerly]", industry had "become productive in the compound ratio of their numbers and their skill" and in every civilized society an increasingly smaller portion of the population was then sufficient to feed everyone (pp. 1269–1270). As he put it, the Malthusian doctrine was by then so discredited that "nobody, except a few mere writers, now troubles himself about Malthus on Population", although these errors may yet "linger in the Universities, the appropriate depositories of what is obsolete and practically unimportant" (p. 1270).

Sadly, many later university professors and social activists—including some early twentieth century eugenicists discussed without context in *Growth*—proved very apt at promoting the Malthusian perspective and creating a constant stream of new catastrophist scenarios [27,94]. When their teachings prevailed, human suffering followed, from forced sterilizations to wars of aggression. Although one is reluctant to bring this up, Professor Smil, who was born under Nazi occupation, shouldn't need to be reminded that Hitler justified his policy of territorial expansion on Malthusian grounds [95]. Yet, as the reviewer of popular 1948 neo-Malthusian tracts observed in Time magazine, how could anyone who

bought into the belief that national governments had too few resources to keep their populations passably well fed argue against the notion that they should ultimately conquer and clear other lands of their populations? What especially saddened the reviewer was that Germany, a country that had managed to "stretch" the sandy acres of the Prussian plain through innovative farming practices and highly skilled industry, had by then already gone to war twice because of the prevalence of the "slice-of-cake [that can't be grown] philosophy" among its people and political leadership [96].

Needless to say, Professor Smil's ire never translates into anything remotely resembling old-fashioned Geopolitik. His targets are rather Angry Birds and "other inane apps" [11] (p. xiv), "unneeded junk" (p. xvi); "poorly built, odd looking, and esthetically offensive" McMansions (p. 251); desires to "out-American America in ostentatious consumption" (p. 501); and failure to appreciate natural beauty. Yet, one can easily imagine how Smil's thinking, like that of many other Malthusians before him, might motivate less sophisticated minds to oppose lifting people out of poverty and promote coercive population control.

Fortunately, basic numbers don't lie and what they convey is that, in the context of market economies, past Promethean writers proved much more right than their opponents. This is not to say that environmental challenges are non-existent, but that the root causes of most current problems are arguably more ideological and institutional than physical. Like nearly all prominent neo-Malthusians before him, however, in *Growth* Professor Smil elected not to directly engage with the key insights of his more optimistic opponents. Knowing his work ethic, then, would it be too much to ask him to consider devoting one of his future book projects to a more systematic debunking of the ideas of people whose predecessors proved much more correct than those on his side of the sustainable development divide? Most important of all, Professor Smil could back up his case with hard numbers rather than having to rely on scenarios and models based on historically debunked premises.

**Funding:** This research received not external funding.

**Conflicts of Interest:** The author declares no conflict of interest.

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
