# Peer review of "The Paradoxical Malthusian. A Promethean Perspective on Vaclav Smil’s Growth: From Microorganisms to Megacities (MIT Press, 2019) and Energy and Civilization: A History (MIT Press, 2017)"

_energies, doi:10.3390/en13205306_

Round 1

Reviewer 1 Report

Dear Author,

thank you for the opportunity to review the manuscript. I have read it with interest. Overall, I share the Author's critical view on modern (neo-Malthusian) environmentalism. The paper is valuable, but there are some minor revisions to be considered/done.

1.The purpose of the study is clearly presented in the abstract. The main goal of the paper, however,  is presented not clearly enough in the Introduction.

  1. Introduction: Please give a brief explanation and justification, why the Author's critical discussion of Smil's key neo-Malthusian assertions was based on the work of past writers (XIX/XX century). It was mentioned on the page 10. In my opinion, the Author should explain this issue earlier.
  2. Section 2.2, p.6: " To the extent that Smil discusses the economic way of thinking critically, it is mostly limited to questionable concepts such as GDP statistics, approaches and models that treated innovation as an exogenous factor, and a failure to appreciate the fundamental role of energy in economic activities". In order to support Author's view, it is worth noting that limitations of GDP as a welfare measure are widely recognized and discussed by economist. Moreover, endogenous growth theory is of crucial importance in economics. Critical thinking is a part of economic discussion.
  3. Section 2.2, p.8: "...most environmental indicators in societies that have experienced significant economic growth have, with the exception of carbon dioxide emissions, long shown (often significant) improvement over time". It would be interesting from the reader's point of view to present in table some examples of environmental indicators improvements.
  4. Section 3.3. "To work properly, however, prices must be embedded in private property rights and voluntary transactions......". While discussing advantages of market economy, the Author's argumentation would be more convincing, if the Author took also into account the existence of market failure which distort the market price system. E.g. behavioural economics deals with market inefficiencies such as mis-pricing, non-rational decision making, dynamic inconsistency in intertemporal choice.
  5. The summary seems to be vague and the findings inconclusive. In order to make the conclusions more convincing, the Author could refer to Smil's mistaken assumptions, which were mentioned in the introduction (abstract). This would link the beginning and the end of an essay. I leave this argument under Author's consideration.

Author Response

Reviewer 1

thank you for the opportunity to review the manuscript. I have read it with interest. Overall, I share the Author's critical view on modern (neo-Malthusian) environmentalism. The paper is valuable, but there are some minor revisions to be considered/done.

1.The purpose of the study is clearly presented in the abstract. The main goal of the paper, however,  is presented not clearly enough in the Introduction.

- The abstract was completely rewritten and made much more explicit. The introduction was shortened, rewritten and (hopefully) made much more explicit.

Introduction: Please give a brief explanation and justification, why the Author's critical discussion of Smil's key neo-Malthusian assertions was based on the work of past writers (XIX/XX century). It was mentioned on the page 10. In my opinion, the Author should explain this issue earlier.

  • Following the reviewer’s suggestion, I discuss my rationale for doing so in both the abstract and the introduction. I also provide additional thoughts on the subject at the beginning of section 3.

Section 2.2, p.6: " To the extent that Smil discusses the economic way of thinking critically, it is mostly limited to questionable concepts such as GDP statistics, approaches and models that treated innovation as an exogenous factor, and a failure to appreciate the fundamental role of energy in economic activities". In order to support Author's view, it is worth noting that limitations of GDP as a welfare measure are widely recognized and discussed by economist. Moreover, endogenous growth theory is of crucial importance in economics. Critical thinking is a part of economic discussion.

  • I actually agree with the referee and obviously didn’t express myself clearly. I have rewritten this passage accordingly (section 2.2)

Section 2.2, p.8: "...most environmental indicators in societies that have experienced significant economic growth have, with the exception of carbon dioxide emissions, long shown (often significant) improvement over time". It would be interesting from the reader's point of view to present in table some examples of environmental indicators improvements.

  • The text is already lengthy (over 14,000 words including nearly 100 references). I hope the referee will forgive me for not having acted upon this suggestion. Apart from specific references, I have referred the reader to the website Our World in Data which contains the most detailed and user-friendly collections of such graphs and tables.

Section 3.3. "To work properly, however, prices must be embedded in private property rights and voluntary transactions......". While discussing advantages of market economy, the Author's argumentation would be more convincing, if the Author took also into account the existence of market failure which distort the market price system. E.g. behavioural economics deals with market inefficiencies such as mis-pricing, non-rational decision making, dynamic inconsistency in intertemporal choice.

  • I have tried to address the author’s basic concerns in section 3.3., but somewhat implicitly and without using much economic jargon or recent technical work. Based on past experience, my (perhaps vain) hope is that the text might prove of interest to a somewhat broader audience that is often hostile to technical economic work and jargon, whether it makes a case they approve of or not.

The summary seems to be vague and the findings inconclusive. In order to make the conclusions more convincing, the Author could refer to Smil's mistaken assumptions, which were mentioned in the introduction (abstract). This would link the beginning and the end of an essay. I leave this argument under Author's consideration.

  • The abstract, introduction and conclusion were completely rewritten in an attempt to address the problems raised by the reviewer.

Reviewer 2 Report

This is a timely paper based on a comparison of Vaclav Smil’s two books:
Growth: From Microorganisms to Megacities (MIT Press, 2019) and Energy and
Civilization: A History (MIT Press, 2017). 

The author has done a great work but before recommending this paper for publication I would like to suggest the author to add several lines in conclusions on how other authors could further debate the ideas presented in this paper. Also, I think that in the introduction section the author has to make more clear the aims of the paper and to better highlight the international importance of this critical study.

Author Response

This is a timely paper based on a comparison of Vaclav Smil’s two books: Growth: From Microorganisms to Megacities (MIT Press, 2019) and Energy and Civilization: A History (MIT Press, 2017). 

The author has done a great work but before recommending this paper for publication I would like to suggest the author to add several lines in conclusions on how other authors could further debate the ideas presented in this paper. Also, I think that in the introduction section the author has to make more clear the aims of the paper and to better highlight the international importance of this critical study.

  • The abstract, introduction and conclusion were completely rewritten in an attempt to address the reviewer’s concerns.
  •  
  • In the conclusion I use a reference to geographer W. B. Meyer’s recent book on the intellectual history of left-wing Prometheanism to allude to the fact that a more detailed history of right-wing (or free-market) Prometheanism between Malthus and the Post-World War II era has yet to be written. Many authors have already addressed some pieces of the puzzle, but I believe a broader synthesis is still warranted and would contribute something original to this long-standing debate. I also made a specific suggestion to Professor Smil...